# Single Ultrasonic-Assisted Washing for Eco-Efficient Production of Mackerel (*Auxis thazard*) Surimi

**DOI:** 10.3390/foods12203817

**Published:** 2023-10-18

**Authors:** Worawan Panpipat, Pornthip Tongkam, Hasene Keskin Çavdar, Manat Chaijan

**Affiliations:** 1Food Technology and Innovation Research Center of Excellence, School of Agricultural Technology and Food Industry, Walailak University, Nakhon Si Thammarat 80160, Thailand; pworawan@wu.ac.th (W.P.); pohntip23@gmail.com (P.T.); 2Department of Food Engineering, Faculty of Engineering, Gaziantep University, University Boulevard, TR-27310 Gaziantep, Turkey; hasenekeskin@gantep.edu.tr

**Keywords:** mackerel, surimi, washing, ultrasonication, gel

## Abstract

This study highlights a promising single washing method for producing dark-fleshed mackerel surimi aided by ultrasonication in conjunction with cold carbonated water containing 0.6% NaCl and mixed antioxidants (0.5% EDTA/0.2% sodium erythorbate/0.2% sodium tripolyphosphate) (CSA). Different washing periods (5, 10, and 15 min) with and without ultrasound were tested. Unwashed mince (A1) and conventional water-washed surimi (10 min/cycle, 3 cycles) (A2) were used as controls. A3, A4, and A5 were subjected to ultrasound-assisted washing for 5, 10, and 15 min, respectively, whereas A6, A7, and A8 had non-ultrasound-assisted washing for 5, 10, and 15 min. Results showed that the surimi yield decreased as the ultrasonic treatment time increased from 5 to 15 min (*p* < 0.05). Increased ultrasonic time resulted in greater protein denaturation, protein oxidation, myoglobin removal, and lipid oxidation in surimi (*p* < 0.05). Surimi produced by CSA ultrasonication for 5 min (A3), on the other hand, had a comparable overall quality to A2 surimi (*p* > 0.05). The correspondence gel (A3) outperformed the control gel (A2) in terms of gel strength, whiteness, and water-holding capacity (*p* < 0.05). The formation of regularly continuous, more organized, and smooth network structures in surimi gel was observed in A2 and A3 gels, whereas sparse and larger pore sizes were noticed in surimi gels produced by longer ultrasonic treatment. All of the surimi gels had identical FTIR spectra, indicating that the functional groups of the protein gel were consistent throughout. As a result, a single 5 min CSA-ultrasonic washing could potentially yield surimi of comparable quality to conventional washing. This could pave the way for the development of dark-fleshed fish surimi, which would require less washing time and produce less waste water.

## 1. Introduction

Surimi is a concentrated myofibrillar protein obtained by washing fish mince to remove non-gel-forming proteins and myoglobin [1]. Surimi’s raw materials are primarily white fish; however, owing to overfishing and environmental changes, white fish are becoming insufficient to supply the market for surimi production [2]. Although dark-fleshed fish can be used as an alternative raw material for surimi production, the abundance of non-gel-forming proteins with high levels of myoglobin and lipids has made it more difficult. This made producing high-quality surimi difficult [2]. Furthermore, dark muscle has greater protease activity than ordinary muscle [3]. This may end up affecting the gel weakening which happens when the gel is maintained at 50–60 °C for a long duration [4]. Lipid oxidation mediated by heme proteins, particularly myoglobin, appears to be an additional concern in surimi produced from some dark-fleshed fish [5]. Lipid oxidation, which results in unpleasant flavors, protein denaturation, and diminished gelling ability of fish muscle via free radical production, may be one of the primary variables affecting storage life. It was critical to use an effective washing strategy to remove as many unwanted compounds as possible from the mince of dark-fleshed fish. Surimi is typically produced by washing mechanically separated fish mince three times in cold water (5–10 °C) until most of the water-soluble components have been removed. Surimi contains a high concentration of myofibrillar proteins, which primarily contribute to gel development [4]. The traditional washing method may not be sufficient to separate non-gel-forming proteins, lipids, and myoglobin from dark fish mince; thus, novel washing methods are still required. Another consideration for making a sustainable and environmentally friendly process is to reduce the amount of freshwater used during surimi washing.

The number of washing cycles required during surimi production is determined by the species, type of wash, condition, and quality of the final surimi [6]. Approximately 15 kg of water is used in the processing of 1 kg of surimi [7], resulting in a negative environmental impact caused by the processing water. Therefore, it is essential to produce surimi in an environmentally friendly manner with fewer washing cycles in order to minimize waste, maximize yield, as well as decrease wash water volume. Previous research took this into account and found that each unique washing medium with a rising number of cycles is efficient for enhancing the quality aspects of surimi. Alkaline saline washing in sutchi catfish [7] and silver carp [8] was investigated with three washing cycles. In a previous study [9], we used cold carbonated water (CW) for the production of mackerel surimi. To the best of our knowledge, there have been few reports on ultrasonic aided washing (UAW)’s introduction of a single washing cycle for the production of dark-fleshed fish surimi. A novel washing media, specifically CW with antioxidants, was used to maximize the removal of unpleasant compounds [10]. Since ultrasonication can cause strong cavitation [11], which can contribute to the chemical degradation of proteins and lipids during washing, the inclusion of antioxidants in washing media may alleviate this issue.

A more efficient washing process for surimi production must be considered in order to accomplish the sustainable and successful development of dark-meat fish surimi manufacturing. However, there have been few studies on the effect of UAW combined with carbonated water-mixed antioxidant media on the quality of mackerel surimi. In light of this, we investigated the effect of reduced washing cycles on surimi quality, as well as the effect of UAW at various times on the physicochemical characteristics and gel-forming capability of mackerel surimi, in order to serve as a theoretical guide for the least water-intensive manufacturing of high-quality surimi from dark-fleshed fish.

## 2. Materials and Methods

### 2.1. Chemicals

All chemicals used in the analysis, such as sodium tripolyphosphate, sodium erythorbate, sodium nitrite, bathophenanthroline, potassium iodide, and cholroform, were obtained from Sigma-Aldrich Corp. (St. Louis, MO, USA).

### 2.2. Surimi Preparation by UAW

Fresh frigate mackerel (*A. thazard*) with an average weight of 100–120 g were purchased from the local Thasala market approximately 12 h after capture. Then, the fish were transferred in ice with an ice-to-fish ratio of 2:1 (*w*/*w*) to Walailak University’s Food Technology and Innovation Research Center of Excellence within 20 min. The fish were immediately washed in cold tap water (4 °C). The gutted, filleted, skinned, and entire muscles were gathered. Fish mince was produced by grinding fish fillets in a meat grinder (a Panasonic MK-G20MR, Tokyo, Japan) with a 4 mm hole diameter. 

To prepare surimi, different washing periods (5, 10, and 15 min/cycle) with and without ultrasound were tested in this study using a single washing cycle of CW containing 0.6% NaCl and mixed antioxidants (0.2% sodium tripolyphosphate/0.2% sodium erythorbate/0.5% EDTA) (CSA) [10]. A medium-to-mince ratio of 3 to 1 (*v*/*w*) was used for washing. Unwashed mince (A1) and conventional water-washed surimi (10 min/cycle, 3 cycles) (A2) were used as controls. A3, A4, and A5 had ultrasound-assisted washing for 5, 10, and 15 min, respectively, whereas A6, A7, and A8 had non-ultrasound-assisted washing for 5, 10, and 15 min (Table 1). An ultrasound treatment was carried out using an ultrasonicator (Sonics, Model VC750, Sonica & Materials, Inc., Newtown, CT, USA). A flat-tip probe with a diameter of 25 mm was employed with an intensity of 153 W/cm^2^ at a single frequency of 20 kHz and an amplitude of 80%. After being washed, mince was dewatered by being passed through a layer of nylon screen, then hydraulically pressed to achieve a final moisture content of about 80%. The yield of surimi from various washing practices was calculated based on the weight of surimi in relation to the weight of minced fish. Each sample was well combined with 4% (*w*/*w*) sucrose and 4% (*w*/*w*) sorbitol before being frozen in an air blast freezer (Polar DN494 367 Blast Freezer, Campbell town, NSW, Australia). The frozen samples were stored at −18 °C until they were analyzed. Surimi underwent extensive testing for physicochemical properties and gel-forming ability.

### 2.3. Analyses of Surimi Characteristics

#### 2.3.1. pH 

The sample was homogenized with 10 vol of deionized water (*w*/*v*) in a homogenizer (IKA^®^, Model T25 digital ULTRA-TURRAX^®^, Staufen, Germany), and the pH was analyzed with a calibrated pH meter (Cyberscan 500, Singapore) [12]. 

#### 2.3.2. Lipid Content 

The lipid was extracted using the Bligh and Dyer method [13] with a 50:100:50 mixture of chloroform, methanol, and distilled water. The lipid content was calculated and reported as a g/100 g sample.

#### 2.3.3. Heme Iron and Non-Heme Iron Contents

The heme iron was determined as described by Benjakul and Bauer [14], using direct spectrophotometric measurement at 525 nm. Heme iron was computed using myoglobin, which contains 0.35% iron [15], and recorded in a mg/100 g sample.

The non-heme iron was determined by the method of Rhee and Ziprin [16] and the content was expressed as a mg/100 g sample.

#### 2.3.4. Absorption Spectra of Myoglobin, Myoglobin Content, and Myoglobin Derivatives

A Shimadzu UV-2100 spectrophotometer (Shimadzu Scientific Instruments Inc., Columbia, MD, USA) was used to measure the myoglobin content at 525 nm. The myoglobin concentration was estimated using a millimolar extinction coefficient of 7.6 with a molecular weight of 16,110 [13] and measured in a g/100 g sample. 

The absorbance (A) was also measured at 503, 525, 550, 557, 582, and 630 nm, and the proportions (%) of different redox stages of myoglobin were estimated using modified Krzywicki’s equations [17] as follows:[Deoxymyoglobin] = −0.543R1 + 1.594R2 + 0.552R3 − 1.329(1)
[Oxymyoglobin] = 0.722R1 − 1.432R2 − 1.659R3 + 2.599(2)
[Metmyoglobin] = −0.159R1 − 0.085R2 + 1.262R3 − 0.520(3)
where R1 = A582/A525, R2 = A557/A525, and R3 = A503/A525. 

The absorbances at 582, 525, 557, and 503 nm are designated as A582, A525, A557, and A503, respectively.

#### 2.3.5. Trichloro Acetic Acid (TCA) Soluble Peptide 

Two grams of finely chopped samples were homogenized with 18 mL of 5% TCA for 2 min at a speed of 11,000 rpm using an IKA^®^ homogenizer. The homogenate was incubated at 4 °C for 1 h before being centrifuged at 8000× *g* for 5 min (Sorvall, Model Heraeus Biofuge Stratos, Hanau, Germany). The concentration of TCA-soluble peptides in the supernatant was analyzed by the Lowry assay [18] and expressed as a μmole tyrosine/g sample [19].

#### 2.3.6. Ca^2+^-ATPase Activity

The Ca^2+^-ATPase activity of natural actomyosin (NAM) from unwashed mince or surimi was determined using the approach of Benjakul et al. [12]. The activity was reported as µmoles inorganic phosphate released (Pi)/mg protein/min. 

#### 2.3.7. Reactive Sulfhydryl (SH) Content

The reactive SH content was measured using 5,5′-dithiobis (2-nitrobenzoic acid) (DTNB) according to the method of Ellman [20]. The SH content was estimated using the molar extinction of 13,600 M^−1^ cm^−1^ and reported as mol/10^8^ g protein.

#### 2.3.8. Surface Hydrophobicity 

The hydrophobicity of the sample was determined using bromophenol blue (BPB) [21]. The absorbance of the supernatant was measured at 595 nm against a blank of phosphate buffer. The surface hydrophobicity was expressed as the content of BPB bound.
BPB bound (µg) = 200 µg × (A control − A sample)/A control(4)
where A = absorbance at 595 nm

#### 2.3.9. Rheological Property 

According to Somjid et al. [22], unwashed mince and surimi pastes were evaluated using a HAAKE MARS 60 Rheometer (Thermo Fisher Scientific Inc., Yokohama, Japan) from 10 to 90 °C at an average speed of 2 °C/min for rheological parameters (elastic modulus, G′; viscous modulus, G″; and tan, δ).

### 2.4. Determination of Surimi Gel Functionalities

#### 2.4.1. Gel Formation

To make the gels, thawed surimi or unwashed mince samples (core temperature ~0 °C/moisture content ~80%) were then cut into small pieces. Dry NaCl (2.5% *w*/*w*) was added to the samples and chopped for 5 min to obtain the sol. Then, the sol was filled into a polyvinylidine casing (2.5 cm diameter) and tightly sealed on both ends. The sol was subsequently kept at 40 °C for 30 min before being cooked at 90 °C for 20 min [23,24]. Prior to analysis, the gels were chilled in ice water and kept at 4 °C for 24 h.

#### 2.4.2. Determination of Whiteness

The colorimetric values of gels with the same weight were obtained using a portable Hunter Assoc. Laboratory Miniscan/EX instrument (10° standard observers, illuminant D65, Reston, VA, USA). The *L** (lightness), *a** (redness/greenness), and *b** (yellowness/blueness) were recorded. The whiteness was calculated as follows:Whiteness = 100 − [(100 – *L**)^2^ + *a**^2^ + *b**^2^]^1/2^(5)

#### 2.4.3. Texture Analysis

The breaking force (gel strength) and deformation (elasticity/deformability) of the gels were measured using a TA-XT2 texture analyzer (Stable Micro Systems, Godalming, Surrey, UK) outfitted with a spherical plunger (diameter 5 mm; depression speed 60 mm/min). The test was performed at room temperature (28–30 °C). Three cylindrical-shaped samples with a length of 2.5 cm were used [24].

#### 2.4.4. Determination of Expressible Moisture

A 0.5 cm thick gel sample was weighed and sandwiched between two pieces of Whatman filter paper No. 1 on top and three pieces of the same filter paper on the bottom. The standard mass (5 kg) was placed on top of the sample and held in place for 2 min. After that, the sample was removed and weighed again. The expressible moisture was calculated as a percentage of the sample weight [24].

#### 2.4.5. Lipid Oxidation

The peroxide value (PV) was determined using Botsoglou et al.’s method [25] and reported as milliequivalents of free iodine per kg of lipid. Conjugated diene (CD) was determined using Frankel et al.’s method [26] and reported as the absorbance at 234 nm. Thiobarbituric acid-reactive substances (TBARS) assay was carried out according to Buege and Aust [27] and reported as the mg malondialdehyde (MDA) equivalent per kg of sample.

#### 2.4.6. Evaluation of Rancid and Fishy Odors 

Ten participants (20–40 years old) tested the gel’s sensory properties. All of the panelists are researchers and graduate students who conduct research on fish and seafood with adequate skills in sensory evaluation of fish and surimi products. The panelists regularly ate surimi and had never experienced any surimi allergies. The panelists were recruited for their expertise in assessing the flavor of cooked surimi gel as well as their sensitivity to fishy and rancid odors. They were all trained to have a thorough understanding of rancid and fishy odors [10]. Fishy odor and rancid odor intensity were graded on a 5-point scale (0 = none/4 = strong) [28]. The Walailak University Human Research Ethics Committee approved the protocol for the study (WUEC-21-125-02).

#### 2.4.7. FTIR Spectra 

The method used by Somjid et al. [23] was employed to measure the FTIR. The spectra, which were acquired in 16 scans at a resolution of 4 cm^−1^, were in the 4000–400 cm^−1^ range with automatic signal gain and were compared to a background spectrum taken from the clear, empty cell at 25 °C.

#### 2.4.8. Determination of Microstructure of Gel 

A scanning electron microscope (SEM) (GeminiSEM, Carl Ziess Microscopy, Oberkochen, Germany) was used to examine the microstructures of gels at a 10 kV acceleration voltage [22].

### 2.5. Statistical Analysis

All of the measurements were run in triplicate (*n* = 3), with the exception of the sensory assessment, which was obtained from a panel of 10 participants. An ANOVA was performed on the data. Duncan’s multiple range analysis was used to compare the means. The statistical analysis was performed with SPSS 23.0 (SPSS Inc., Chicago, IL, USA).

## 3. Results and Discussion

### 3.1. Yield and Chemical Characteristics

Table 2 compares the yield and chemical properties of mackerel surimi produced by a single washing with a cold cocktail carbonate-saline-antioxidant (CSA) solution aided by ultrasonication (UW) to CSA washing alone and conventional surimi washing. All three CSA-assisted UWs (A3–A5) yielded less surimi than the traditional washing method (A2) and CSA alone (A6–A8). The increased sonication time was associated with lower surimi yield in the UW group (Table 2). The strong cavitation caused by the UW partially disrupted the polypeptide chains, making them more water soluble and resulting in greater loss of myofibrillar protein with washing water [22]. The highest surimi yield was observed in a single washing with CSA for 5 min (A6 for 80.26%, *p* < 0.05), followed by washing with CSA for 10 and 15 min, which was higher than conventional surimi washing (A2). This could be because a single washing reduced the loss of myofibrillar protein from mackerel mince compared to three-cycle conventional surimi washing. 

All surimi had a pH ranging from 5.87 to 6.02, which was slightly higher than the mackerel mince (pH 5.52). The presence of EDTA, sodium erythorbate, and sodium tripolyphosphate in the washing medium is primarily responsible for the slight increase in the pH of surimi [10]. Mackerel surimi produced with a single CSA assisted by UW had a higher potential to remove lipids from the mince than a CSA washing without UW (*p* < 0.05). The remaining lipids from the UW-produced group were remarkably comparable to the surimi produced by the three-cycle regular washing (A2) (Table 2). This finding suggested that using ultrasonication could improve lipid removal from high-lipid mackerel for the production of high-quality surimi with a single washing means. It should be noted that increasing the ultrasonication time from 5 to 15 min had no effect on the residue lipids in the surimi (*p* > 0.05). 

Furthermore, surimi treated with CSA–UW contained less heme and non-heme iron than surimi washed with CSA alone (*p* < 0.05), indicating that UW has a greater potential to facilitate the removal of both bound and free irons. The single UW was comparable or even better than the three-cycle traditional washing in removing protein-bound and free irons from surimi, which could be reduced by approximately 2–3.5 fold in mackerel mince. According to the findings, while UW provided the lowest surimi yield, the surimi obtained was of superior quality. The use of UW for 5 min could potentially improve surimi quality incomparably or even more than traditional washing, which is more cost effective and produces less waste water.

### 3.2. Myoglobin Content and Changes in Myoglobin Redox State

The whiteness of surimi is one of the most important factors in surimi grading [29]. Myoglobin, which is associated with the whiteness of surimi, is abundant in dark fish muscle like mackerel [28]. Table 3 shows the efficacy of various washing methods in removing myoglobin from mackerel mince. Surimi produced by the CSA–UW group had a lower myoglobin content than CSA alone (*p* < 0.05), which was comparable to the three-cycle regular washing method. There was no difference in myoglobin content between CSA–UW-produced surimi (*p* > 0.05), indicating that applying UW for more than 5 min had a negligible effect on myoglobin removal. The surimi prepared by the CSA–UW group could reduce myoglobin by approximately 3.20 fold from mackerel mince (Table 3). The CSA–UW surimi had higher oxidized metmyoglobin and lower deoxymyoglobin/oxymyoglobin than the regular control washing (*p* < 0.05). This could be due to the strong cavitation force, combined with the CSA, stimulating myoglobin oxidation in the UW-treated group. The induced myoglobin oxidation at a pH of 5–6 may have contributed to the formation of high metmyoglobin in the CSA–UW and CSA surimi groups [9]. The decreased oxymyoglobin in each surimi was related to increased deoxymyoglobin and metmyoglobin, indicating a change in the oxidative state of myoglobin derivative during surimi production. Surimi had higher concentrations of deoxymyoglobin and metmyoglobin while having a lower level of oxymyoglobin when compared to mackerel mince (Table 3). Typically, the presence of different states of myoglobin was associated with the color of surimi [30]; thus, the washing procedure was significant in changing the color of mackerel surimi.

### 3.3. TCA-Soluble Peptides and Protein Denaturation 

Surimi contained less TCA-soluble peptide than mackerel mince due to the loss of indigenous peptides in the washing medium (Table 4). Reduced peptide content also confirmed the establishment of a gel network via protein–protein and protein–water interactions, resulting in lower hydrolysis by digestive enzymes [31]. Unwashed mince (A1) had the highest TCA-soluble peptide level (*p* < 0.05). The soluble peptide was removed from the fish mince using a single washing CSA–UW for 5 min (A3), similar to the surimi prepared by the traditional three-cycle washing method (A2) (Table 4). This demonstrated that using single UW washing provided comparable efficiency in removing peptides or formation of new peptides from mince to the standard three-cycle method. It should be noted that the TCA-soluble peptides in the CSA–UW surimi were lower than in the CSA alone (*p* < 0.05). This could be because the high cavitation force in UW-treated samples altered the intrinsic protease activity, resulting in lower protein hydrolysis. 

Protein denaturation affects surimi gel strength, which is an important parameter for surimi grading. Ca^2+^-ATPase, which is primarily found in the myosin head, is a reliable marker of myosin denaturation and is necessary for muscle arrangement, integrity, and functionality [32]. Surimi from CSA–UW and CSA alone had higher Ca^2+^-ATPase activity than the control washing (A2) but lower than mackerel mince (*p* < 0.05). This result indicated the partial denaturation of myosin caused by surimi washing. Furthermore, higher Ca^2+^-ATPase activity was associated with lower protein denaturation in surimi from CSA–UW and CSA alone (Table 4). Polypeptide structural changes occurred to varying degrees after being subjected to physical force during stirring or UW. Increased ultrasonication time significantly reduced Ca^2+^-ATPase activity of surimi (*p* < 0.05), indicating more protein denaturation caused by high cavitation force. It was in line with Tang and Yongsawatdigul [33] and Somjid et al. [22], who found that the Ca^2+^-ATPase activity of myosin exposed to high-intensity ultrasound decreased as the ultrasonic power and time increased. 

Surimi from the CSA–UW group had a higher reactive sulfhydryl (SH) content than mackerel mince (A1), which was comparable to surimi from the CSA group and the regular washing method (A2). It should be noted that the longer the washing period, either for CSA or CSA–UW, the higher the SH content. Gülseren et al. [34] reported that the decrease in SH content could be attributed to SH group oxidation forming disulfide bonds caused by cavitation, which produces some temporal OH and H radicals during washing. The presence of mixed antioxidants prevented protein oxidation during ultrasonication [10], which could explain a comparable reactive sulfhydryl content between CSA–UW surimi, CSA alone, and the control washing method (Table 4). The results confirmed that all washing methods resulted in partial protein unfolding with longer washing times or washing cycles exhibiting a higher degree of protein denaturation with highly exposed SH groups (Table 3). However, Chandrapala et al. [35] observed that ultrasound treatment had no impact on the free SH group content of whey protein concentrate. 

Surimi produced by CSA–UW and CSA alone had higher surface hydrophobicity than traditional washing (*p* < 0.05), except when CSA–UW was used for 15 min. The higher the UW, the lower the surface hydrophobicity of surimi as a result of severe physical forces resulting in exposed hydrophobic regions bound together by hydrophobic interaction [22]. The net surface hydrophobicity of surimi was governed by the degree of protein unfolding and the rate at which hydrophobic interactions formed. This was consistent with Somjid et al. [22], who found that ultrasonication-assisted washing increased the surface hydrophobicity of mackerel surimi compared to conventional washing. Typically, gelation of myofibrillar protein requires proper protein unfolding, which progresses through cross-linking to form gel networks [36]. According to the findings, implementing CSA–UW for 5 min could potentially preserve the protein in comparable or even better than three times traditional washing, resulting in a 3-fold reduction in washing time. 

### 3.4. Lipid Oxidation and Off-Odor Development 

The effect of various washing methods on the lipid oxidation of surimi is displayed in Table 5. The lowest PV was found in surimi produced by CSA for 5 min (A3; *p* < 0.05), while the highest PV was found in surimi produced by CSA-UW for 10 min (A4; *p* < 0.05). The CD value of surimi produced by all single washing methods (CSA or CSA–UW) was lower than that of surimi produced by the three-cycle conventional washing method (*p* < 0.05). The PV and CD of unwashed mince were either lower or higher than that of surimi depending on the washing condition (Table 5). The variation in PV and CD among surimi produced by different washing methods, on the other hand, may be related to residual lipid, physical force, washing time, and the presence of antioxidants in the washing media (Table 5). Furthermore, PV and CD are unstable primary lipid oxidation products that can degrade into secondary lipid oxidation products. As a result, both values may rapidly change during the manufacturing and storage of surimi. It should be noted that the presence of mixed antioxidants in the washing media could effectively prevent lipid oxidation of surimi toward cavitation force and other severe conditions during surimi washing [10]. 

The TBARS of surimi produced by novel single washing methods were slightly higher or comparable to the standard three-cycle washing method, but less than the correspondence mackerel mince (*p* < 0.05). This indicated a lower degree of lipid oxidation, which resulted in a lower lipid content in the surimi. The outcome agreed with Eymard et al. [37], who noticed lower TBARS in surimi when compared to the original horse mackerel mince due to less subcutaneous fat and dark muscle remaining after hand-mincing. The presence of mixed antioxidants could also impede lipid oxidation during washing with CSA or CSA–UW, resulting in a small difference in TBARS after increasing washing time (Table 5). Kang et al. [38] observed ultrasound-induced lipid oxidation in beef during curing as a result of sonolysis. Cavitation can raise the temperature, hastening lipid oxidation [39]. In this study, ultrasound was found to be ineffective in promoting lipid oxidation because the washing was performed at a low temperature. There was no significant difference in rancid odor between all surimi and mackerel mince (*p* > 0.05). This result was strongly related to the small differences in lipid oxidation parameters like PV, CD, and TBARS between treatments (Table 5). 

Because of the low levels of TBARS found in surimi and unwashed mince, the rancid odor was less than one and did not differ between samples (*p* > 0.05). Surimi, on the other hand, had a less fishy odor than the parent mackerel mince (*p* < 0.05). There were small variations in fishy odor among surimi produced using distinct washing methods (Table 5), indicating that a developed single washing method was comparable to the three-cycle traditional washing method in removing fishy odor substances. 

### 3.5. Rheological Properties 

Figure 1 depicts the dynamic thermal gelation profiles of surimi produced by various washing methods in terms of G′, G″, and tan δ from 20 to 90 °C. In general, the G′ represents a sample’s elastic behavior by measuring the deformation energy stored in it during shearing [10,22]. G′ changes during heating were caused by denaturation and cross-linking of various myofibrillar protein components at different temperatures. G′ and G″ remained constant during heating from 20 to 50 °C, then dropped slightly between 50 and 55 °C, indicating partial denaturation of myofibrillar protein as well as residual protease activity [40]. A similar drop in G′ and G″ was also observed in Indian squid (*Loligo duvauceli*) mantle proteins [40]. Denaturation of actin caused an increase in the number of cross-links between protein strands or aggregates, as well as the deposition of extra denatured proteins in the protein networks, strengthening the gel network, at 55–60 °C [41]. G′ and G″ continue to increase as the temperature rises above 60 °C, indicating the formation of a highly elastic myofibrillar protein gel. As the temperature increased to 90 °C, the G′ and G″ values of all samples increased continuously, indicating the formation of the actin-myosin network structure. G′ was greater than G″, indicating that the viscous properties of surimi pastes outweighed their elastic properties (Figure 1a,b), demonstrating typical viscoelastic solid-like gel network behavior. The significantly higher G′ and G″ of control A2 surimi at the start showed protein aggregation. The first protein network structure was constructed using the “gel setting” period, which can be observed in A3–A8. Myosin will unravel at this point to allow for ordered polymerization, and the initial elasticity of proteins will be lost [42]. This spectrum, according to Buamard et al. [43], includes the formation of protein connectivity via weak links among protein molecules, such as hydrogen bonds. The G′ was then increased again, reaching a peak of around 60–65 °C, suggesting the development of a robust gel matrix due to more attractive forces (e.g., hydrophobic and disulfide bonds). G′ then gradually raised until it reached its peak at the end. During the gel reinforcing step, myosin polymerization and initial cross-linking occur. The G′ and G″ pattern was present in all single washing surimi and unbaked mince (Figure 1a,b). Tan δ was less than 1.0 for all samples because G′ values were greater than G″ values (Figure 1c). As a result, all of the samples showed the features of an elastic fluid with the improved gelling ability [44]. The tan δ was similar even though the final G′ and G″ of the surimi varied, implying that all surimi will undergo the same degree of sol–gel transformation after heating. As a consequence, all of the samples can be gelled to varying degrees upon the two-step heating (40 °C/30 min 90 °C/20 min). All surimi washed with CSA or CSA–UW had similar rheological aspects to other dark-fleshed fish surimi; however, the values of G′, G″, and tan δ differed to some extent [22].

### 3.6. Gel Characteristics

The appearance of surimi gels prepared using various washing methods is depicted in Figure 2. Surimi gels were all lighter in color than unwashed mince gel, due to lowered residual myoglobin (Table 3). All samples formed smooth surface gels with a stable tubular shape (Figure 2). Principally, surimi quality is determined by the gel strength, water-holding capacity, and color. The gel prepared by CSA–UW for 5 min produced surimi (A3) with the greatest breaking force (*p* < 0.05), which was comparable to the gel prepared by three-cycle regular-washed surimi (A2). It should be noted that the other CSA–UW and CSA surimi could form a weak gel, as indicated by the lower breaking force than the mackerel mince (Figure 3a). This could be due to the high degree of denaturation that occurred during surimi washing, resulting in protein aggregation with low gel network formation. The elasticity of the gel, as determined by deformation, could confirm the looser gel network formation of surimi prepared from CSA–UW and CSA, with the exception of A2, due to the low deformation (Figure 3b). Typically, the protein gel structure arises by balancing protein–protein and protein–water interactions, resulting in the development of an elastic gel. The low gel elasticity reveals a low water-holding capacity in the gel network. The gel produced from A3 surimi had the greatest deformation (*p* < 0.05), with no statistically significant difference from the control surimi (A2; *p* > 0.05). The amount of entrapped water was proportional to the gel strength and surimi gel deformation [45]. Principally, surimi proteins stretch and denature to form a three-dimensional gel network structure capable of entrapping a large amount of water. Furthermore, using UW for 5 min (A3) may stabilize endogenous transglutaminase (TGase) and inactivate endogenous protease, allowing intermolecular forces to form during the surimi sol setting at 40 °C. Zhang et al. [46] proposed that ultrasonic washing activated endogenous TGase and deactivated endogenous protease during gel formation. Specific functional groups, such as hydrophobic moieties and SH groups, can also promote protein–protein and protein–water interactions [22]. However, washing with an extended ultrasonication time of up to 15 min can reduce the gel strength of the final surimi due to myofibrillar protein denaturation and aggregation, as well as a suppression of TGase activity, resulting in severe protein unfolding as evidenced by high surface hydrophobicity (Table 4), and these aggregates could not form a gel network structure. As a result, using a single CSA–UW for a short period of time (5 min) resulted in high surimi quality, as indicated by high gel strength, being the same as conventional washing surimi.

Color is a significant indication for determining the physical properties of foods and can predict whether the products are likely to be accepted by consumers. Figure 4a depicts the effects of different washing procedures on the whiteness of mackerel surimi gels subjected to UW treatments at different times versus unwashed mince and conventionally prepared surimi. There was no significant difference in the whiteness of surimi gels prepared using different washing methods (*p* > 0.05), which were all higher than unwashed mince gel (*p* < 0.05). This suggested that the ultrasonic treatments had a minor influence on the whiteness of the surimi gel. It should be noted that the new washing procedures could remove myoglobin derivatives in comparison to regular washed surimi (Table 3), resulting in a non-statistical difference in surimi whiteness (Figure 4a). Moreover, surimi gel whiteness is also linked to protein denaturation, aggregation, and gel network formation [47]. Moderate protein denaturation and aggregation result in a dense network of the surimi gel to hold water [45], which changes the plane of light scattering. 

All surimi gels had lower water exudation than the unwashed mince gel (Figure 4b). The presence of ungelled forming proteins in unwashed mince, such as connective tissue and stroma, may inhibit continuous gel network formation, resulting in high water exudation. Typically, protein gels have a honeycomb-like microstructure filled with capillaries, which promotes the retention of native water [48]. The occurrence of a discontinuous network may result in reduced water holding capacity of the gel, resulting in excessive water exudation. The surimi gel produced from CSA–UW for 5 min (A3) lost the least amount of water (*p* < 0.05), outperforming the conventional washing surimi gel (A2). Singh et al. [31] suggested that an ordered gel network structure may be the primary reason for increased water-holding capacity. Among the UW surimi group, increasing the UW time increased the expressible drip from the surimi gel (A3–A5, Figure 4b). This result was associated with lower gel strength of surimi produced by longer UW treatment. Jiao et al. [49] stated that lower water exudation indicates increased water-holding capacity of the protein gels, leading to a higher gel strength and a strong protein network. Moreover, water loss increased with the increasing washing time in surimi gel prepared from CSA (A6–A8). This could be explained by the fact that more protein denaturation occurred when surimi was washed for a longer period of time (Table 4), resulting in a lower water-holding capacity of unfolded protein. 

Changes in washing conditions can cause changes in protein conformation in surimi, such as peptide bond angles and hydrogen bonding, resulting in an increase or decrease in the precise wavenumbers of amide bands responding to FTIR spectra. As a result, FTIR can be used to identify protein structures based on wavenumber shift and peak intensity calculation [50]. Surimi is mostly contained protein in dry weight, with myosin being the most important component in forming the cross-linking protein gel network. The absorption bands in amide I (1655 cm^−1^) and II (1545 cm^−1^) were observed in all surimi gels (Figure 5). Ge et al. [51] reported surimi gels with IR peaks at 1643, 1539, and 1239 cm^−1^ due to the occurrence of amide I (C=O), amide II (N-H), and amide III (C-N) bands. Typically, the IR spectra of surimi gel have very high similarity to raw surimi [52]. Wei et al. [52] noticed that after gelation, the bands of absorption of amides I and II expanded considerably (20%), demonstrating a protein structural change. Hydrogen bond reduction was also observed in surimi gel [53]. Because the protein–protein interaction increased gradually during gelation, hydrogen bonds, ionic linkages, and other weak interactions in proteins are most likely reduced. According to the findings, similar characteristic peaks were observed in all surimi gels, indicating that the washing methods had negligible effects on the overall structure of the myofibrillar protein gel.

The formation of an incomplete fibrillar structure was observed on the rough surface of the unwashed mince gel (A1) with an aggregative mass (Figure 6, A1). The regularly continuous and smooth network structures were noticeable in surimi gel produced by the conventional three-cycle washing method (Figure 6, A2), which could be attributed to the proper gel network formation of myofibrillar protein. In addition, the pores were evenly distributed (Figure 6, A2). Surimi gels, on the other hand, became more sparse and porous after 5 to 15 min of CSA or CSA–UW treatment (Figure 6, A3–A8). The pores in the surimi gel structure continued to grow, but they were not evenly distributed, and the texture was rough and uneven. The use of carbonated water with salt and mixed antioxidants (EDTA and polyphosphate) may result in a high level of protein denaturation (Table 4), as evidenced by the presence of excessive protein aggregates, preventing the development of a homogeneously three-dimensional network. Surimi gels with longer single washing times, both with (Figure 6, A3–A5) and without (Figure 6, A6–A8) ultrasonication, had a rougher structure with larger holes due to excessive protein aggregation. This result was supported by lower gel strength (Figure 3a) and deformation (Figure 3b) with higher expressible drip (Figure 4b) in the CSA and CSA–UW surimi-prepared gels compared to conventional surimi-prepared gel. Larger pores in the gel structure may allow for greater water exudation (Figure 4b). These findings suggest that excessive protein aggregation disrupted the dense homogeneous network of the surimi gels as the duration of the CSA or CSA–UW treatment increased, resulting in the creation of bigger pores [54].

## 4. Conclusions

A single washing assisted by ultrasonication (UW) in conjunction with carbonated water containing salt and mixed antioxidants (CSA) was an innovative process for producing mackerel surimi. Although the surimi yield was lower after washing with UW, the removal of unwanted proteins and overall surimi quality were improved. A longer UW treatment time resulted in more protein denaturation, protein oxidation, and lipid oxidation in surimi. The surimi produced by CSA–UW for 5 min (A3) had comparable or even better overall surimi quality than three-cycle washed surimi. The correspondence gel (A3) exhibited comparable gel strength and whiteness, as well as a high water-holding capacity, to the control gel (A2). This was supported by the formation of regularly continuous and smooth network structures in surimi gel. Thus, a single CSA–UW washing for 5 min could potentially prepare the same quality surimi from dark-fleshed fish muscle as a three-cycle conventional washing method. This could lay the groundwork for the development of surimi produced from dark-fleshed fish, which would require less washing time and produce less waste water.

## Figures and Tables

**Figure 1 foods-12-03817-f001:**
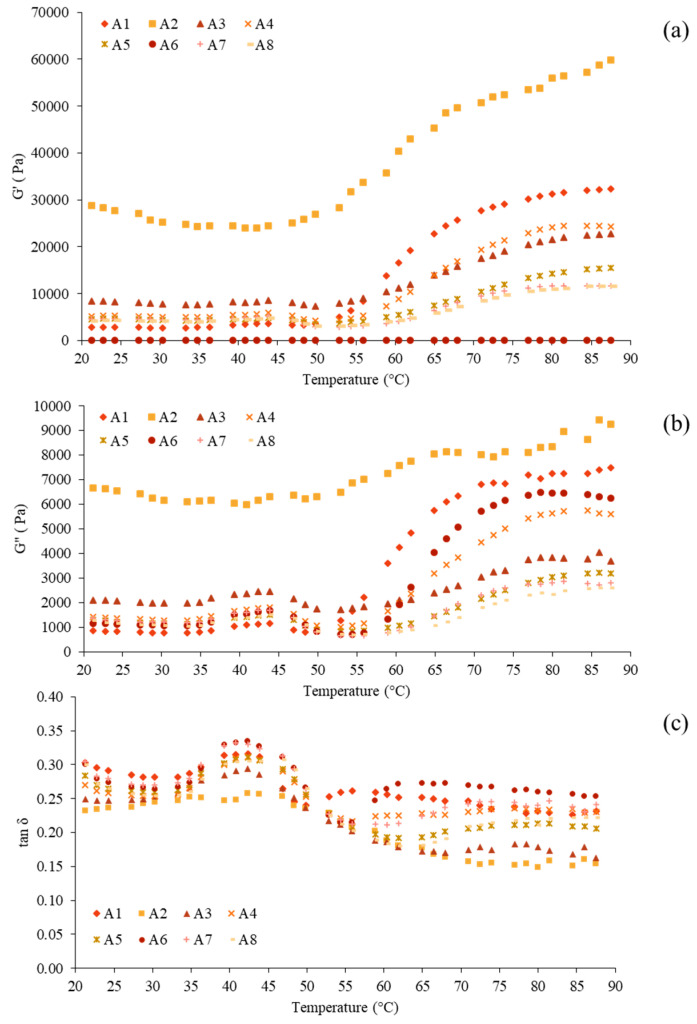
Rheological properties (G′ (**a**), G″ (**b**), and tan δ (**c**)) of mackerel (*Auxis thazard*) unwashed mince and surimi prepared using conventional washing and ultrasonic-assisted single washing processes. Treatment captions can be seen in Table 1 and Table 2.

**Figure 2 foods-12-03817-f002:**
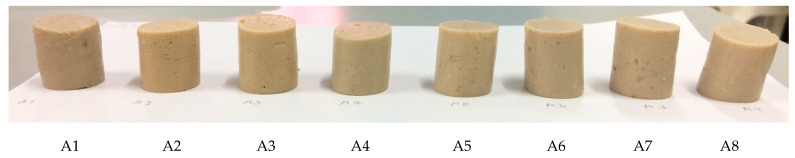
Appearance of mackerel surimi gels prepared using conventional washing and ultrasonic-assisted single washing processes. Treatment captions can be seen in Table 1 and Table 2.

**Figure 3 foods-12-03817-f003:**
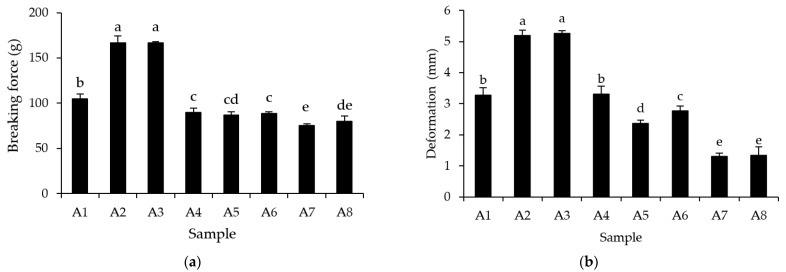
Breaking force (**a**) and deformation (**b**) of gels from mackerel unwashed mince and surimi prepared using conventional washing and ultrasonic-assisted single washing processes. Bars represent the standard deviations from triplicate determinations. Different letters indicate the significant differences (*p* < 0.05). Treatment captions can be seen in Table 1 and Table 2.

**Figure 4 foods-12-03817-f004:**
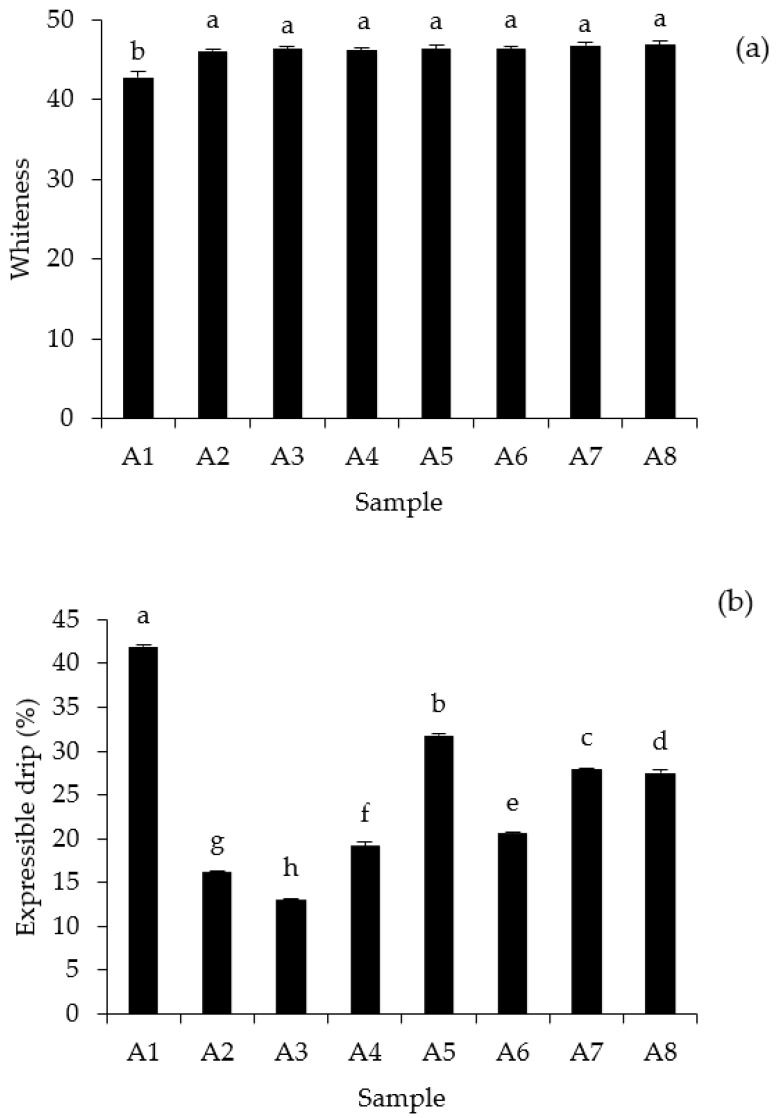
Whiteness (**a**) and expressible drip (**b**) of gels from mackerel unwashed mince and surimi prepared using conventional washing and ultrasonic-assisted single washing processes. Bars represent the standard deviations from triplicate determinations. Different letters indicate the significant differences (*p* < 0.05). Treatment captions can be seen in Table 1 and Table 2.

**Figure 5 foods-12-03817-f005:**
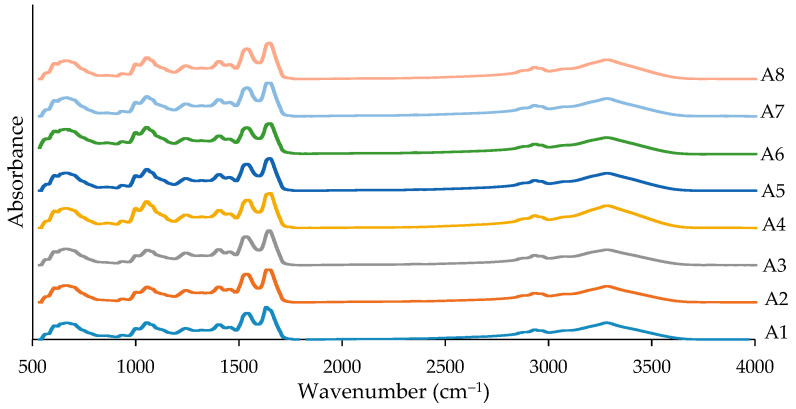
FTIR spectra of gels from mackerel unwashed mince and surimi prepared using conventional washing and ultrasonic-assisted single washing processes. Treatment captions can be seen in Table 1 and Table 2.

**Figure 6 foods-12-03817-f006:**
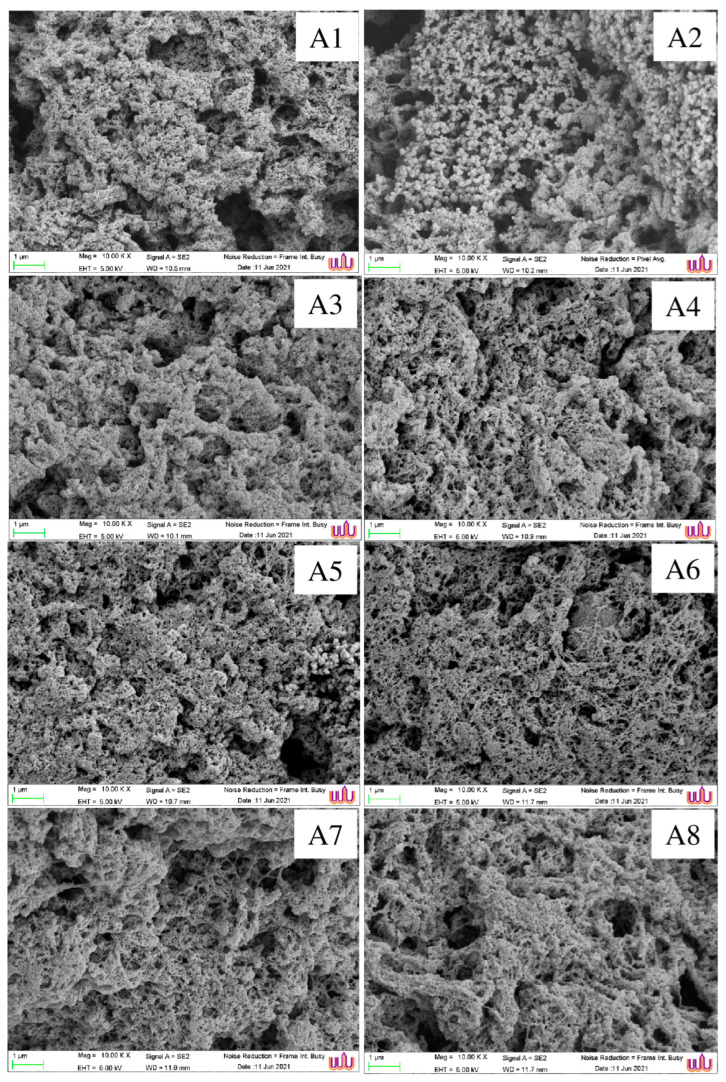
Electron microscopic images of gels from mackerel unwashed mince and surimi prepared with different washing treatments. Magnification: 10,000×, EHT: 5 kV. Treatment captions can be seen in Table 1 and Table 2.

**Table 1 foods-12-03817-t001:** Washing process used in each treatment.

Treatment	1st Cycle	2nd Cycle	3rd Cycle
A1 (unwashed mince)	-	-	-
A2	Water (10 min)	Water (10 min)	Water (10 min)
A3	CSA + US (5 min)	-	-
A4	CSA + US (10 min)	-	-
A5	CSA + US (15 min)	-	-
A6	CSA − US (5 min)	-	-
A7	CSA − US (10 min)	-	-
A8	CSA − US (15 min)	-	-

CSA = cold carbonated water containing 0.6% NaCl and antioxidant (0.5% EDTA/0.2% sodium erythorbate/0.2% sodium tripolyphosphate). + US = with ultrasonication. − US = without ultrasonication.

**Table 2 foods-12-03817-t002:** Yield, pH, lipid content, heme iron content, and non-heme iron content of mackerel unwashed mince and surimi prepared using conventional washing and ultrasonic assisted single washing processes.

Treatment	Yield (%)	pH	Lipid (g/100 g)	Heme Iron (mg/100 g)	Non-Heme Iron (mg/100 g)
A1	-	5.52 ± 0.01g	1.97 ± 0.03a	35.92 ± 0.40a	16.15 ± 0.33a
A2	71.56 ± 1.20c	5.87 ± 0.01f	0.62 ± 0.10de	15.07 ± 0.07d	15.19 ± 0.56b
A3	66.11 ± 1.53d	5.90 ± 0.01e	0.70 ± 0.03e	10.46 ± 0.85f	8.79 ± 0.18cd
A4	65.52 ± 1.60d	5.93 ± 0.01d	0.69 ± 0.03e	15.80 ± 0.12cd	8.38 ± 0.07d
A5	58.10 ± 1.01e	6.00 ± 0.01b	0.64 ± 0.02de	10.43 ± 0.81f	9.39 ± 0.85c
A6	80.26 ± 1.18a	5.96 ± 0.01c	1.50 ± 0.03c	18.45 ± 0.11b	16.85 ± 0.37a
A7	77.62 ± 1.16b	6.02 ± 0.01a	1.77 ± 0.05b	13.38 ± 0.02e	9.21 ± 0.02c
A8	78.14 ± 1.39b	6.01 ± 0.01ab	1.69 ± 0.03b	16.05 ± 0.11c	8.87 ± 0.44cd

Values are presented as the mean ± standard deviation of determinations produced in triplicate. Significant differences (*p* < 0.05) are indicated by different letters in the same row. A1 = unwashed mince. A2 = conventional water-washed surimi (10 min/cycle, 3 cycles). A3–A8 were prepared using a single washing cycle of cold carbonated water containing 0.6% NaCl and antioxidants (0.5% EDTA/0.2% sodium erythorbate/0.2% sodium tripolyphosphate). A3, A4, and A5 = ultrasound-assisted washing for 5, 10, and 15 min, respectively. A6, A7, and A8 = non-ultrasound-assisted washing for 5, 10, and 15 min, respectively.

**Table 3 foods-12-03817-t003:** Myoglobin content and myoglobin redox state of mackerel unwashed mince and surimi prepared using conventional washing and ultrasonic-assisted single washing processes.

Treatment	Myoglobin (g/100 g)	Myoglobin Redox State (%)		
		Deoxymyoglobin	Oxymyoglobin	Metmyoglobin
A1	0.73 ± 0.09a	9.81 ± 1.18e	10.44 ± 2.16b	8.10 ± 1.34a
A2	0.30 ± 0.05cd	25.12 ± 0.14a	12.57 ± 0.89a	2.45 ± 0.04e
A3	0.23 ± 0.02d	17.39 ± 0.50b	8.15 ± 0.69c	4.31 ± 0.24d
A4	0.24 ± 0.00d	12.53 ± 0.31cd	6.39 ± 0.14cd	6.40 ± 0.17bc
A5	0.23 ± 0.02d	13.43 ± 0.52c	8.22 ± 1.52c	5.70 ± 0.23c
A6	0.42 ± 0.03b	9.62 ± 0.84e	5.78 ± 0.74d	8.67 ± 0.89a
A7	0.28 ± 0.0.03cd	11.73 ± 0.03d	6.15 ± 0.95cd	6.99 ± 0.03b
A8	0.31 ± 0.01c	13.09 ± 0.66c	7.28 ± 0.65cd	6.07 ± 0.27bc

Values are presented as the mean ± standard deviation of determinations performed in triplicate. Significant differences (*p* < 0.05) are indicated by different letters in the same row. Treatment caption can be seen in Table 1 and Table 2.

**Table 4 foods-12-03817-t004:** Biochemical properties of mackerel unwashed mince and surimi prepared using conventional washing and ultrasonic-assisted single washing processes.

Treatment	TCA-Soluble Peptide (µmol Tyrosine/g Sample)	Ca^2+^-ATPase Activity (µmol/mg Protein/min)	Reactive Sulfhydryl Content (mol/10^8^ g Protein)	Hydrophobicity (BPB Bound (µg))
A1	1.39 ± 0.12a	1.89 ± 0.06a	2.36 ± 0.01d	52.95 ± 0.20b
A2	0.54 ± 0.01f	0.29 ± 0.04e	4.49 ± 0.17a	18.97 ± 0.41g
A3	0.58 ± 0.05f	1.16 ± 0.21c	4.00 ± 0.17b	38.43 ± 0.78d
A4	0.80 ± 0.04e	0.98 ± 0.03d	4.48 ± 0.06a	50.60 ± 0.36c
A5	0.89 ± 0.04de	0.89 ± 0.05d	4.54 ± 0.13a	13.44 ± 0.36h
A6	0.93 ± 0.05cd	1.26 ± 0.03c	3.54 ± 0.21c	32.15 ± 1.26e
A7	0.97 ± 0.03cd	1.63 ± 0.03b	4.13 ± 0.10b	59.12 ± 0.16a
A8	1.07 ± 0.04b	1.63 ± 0.02b	4.09 ± 0.06b	26.08 ± 0.05f

Values are presented as the mean ± standard deviation of determinations performed in triplicate. Significant differences (*p* < 0.05) are indicated by different letters in the same row. Treatment captions can be seen in Table 1 and Table 2. TCA = trichloroacetic acid. BPB = bromophenol blue.

**Table 5 foods-12-03817-t005:** Lipid oxidation, rancid odor, and fishy odor of gels from mackerel unwashed mince and surimi prepared using conventional washing and ultrasonic assisted single washing processes.

Treatment	Peroxide Value (meq/kg Lipid)	Conjugated Diene (A234)	TBARS (mg MDA Equivalent/kg)	Rancid Odor * ^NS^	Fishy Odor *
A1	1.97 ± 0.11ab	21.22 ± 0.19d	0.48 ± 0.01a	0.60 ± 0.84	2.20 ± 0.92a
A2	0.92 ± 0.03c	67.70 ± 0.79a	0.34 ± 0.01d	0.50 ± 0.71	1.90 ± 0.74ab
A3	0.66 ± 0.01d	15.31 ± 0.36e	0.38 ± 0.02cd	0.70 ± 0.82	1.80 ± 0.79abc
A4	2.22 ± 0.05a	22.01 ± 0.69d	0.38 ± 0.03c	0.30 ± 0.48	1.30 ± 0.67bc
A5	1.70 ± 0.47b	31.32 ± 0.79b	0.43 ± 0.00b	0.40 ± 0.70	1.30 ± 0.82bc
A6	1.84 ± 0.38ab	29.64 ± 0.30c	0.39 ± 0.03bc	0.30 ± 0.67	1.40 ± 0.70bc
A7	1.81 ± 0.16ab	10.61 ± 0.51g	0.48 ± 0.01a	0.20 ± 0.42	1.30 ± 0.67bc
A8	0.95 ± 0.12c	14.04 ± 0.76f	0.48 ± 0.02a	0.10 ± 0.32	1.10 ± 0.57c

Values are given as mean ± standard deviation from triplicate determinations except for off-odor score (*n* = 10). Different letters in the same column indicate significant differences (*p* < 0.05). * A score of 4 represent ‘very strong’ while 0 represents ‘none’. Treatment captions can be seen in Table 1 and Table 2. TBARS = thiobarbituric acid reactive substances. MDA = malondialdehyde. ^NS^ denotes a non-significant difference.

## Data Availability

Data are contained within the article.

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
