# Peer review of "Single Ultrasonic-Assisted Washing for Eco-Efficient Production of Mackerel (Auxis thazard) Surimi"

_foods, 2023, doi:10.3390/foods12203817_

Round 1

Reviewer 1 Report

Comments to the authors
I have completed the evaluation of  manuscript entitled ” Novel single washing approach for eco-efficient production of  mackerel (Auxis thazard) surimi by the aid of ultrasonication”. authors demonstrated that Although the surimi yield was lower after washing with UW, but the  overall surimi quality was improved. Most important finding of this paper may be highlighted that this was aimed to reduce waste water production in surimi plants by washing with unltrasonication . 

Specific points are as below

It is was  felt that  a better title may be thought of to depict work done clearly.

It will be better that for treatments, A1, A2….so on shall be prepared in a table for clear understanding.

The amplitude of the ultrasonication could not found anywhere inn manuscript ? Futher , how did you decide the amplitude for the untrasonication ?

2.4.6 Evaluation of Rancid and Fishy Odors- Ten participants (20–40 years old) tested the gel's sensory properties …..why only ten  participants. It should have been atleast 20 ?

3.5 Rheological Property

Line 407-408 G' and G'' remained  constant during heating from 20 to 50 °C, then dropped slightly between 50 to 55 C, in indicating partial denaturation of myofibillar protein. Similar drop in G’ and G” also reported in squid (Mehta et al., 2017 (https://ifst.onlinelibrary.wiley.com/doi/abs/10.1111/jfpp.12891) . Where they have corroborated this drop or denaturation to catheptic activity also increment at this particular range. However, this range may vary species to species. That can help us to design better cooking temperature for the gel. Hope this is help for the authors. It is advisable to add above discussion and reference to improve the discussion in  this section.  

Figure 5. FTIR spectra of gels from mackerel unwashed mince and surimi prepared 604 using conventional washing and ultrasonic assisted single washing processes.

Bars  represent the standard deviations from triplicate determinations. Different letters 606 indicate the significant differences (p < 0.05). Treatment caption can be seen in Table 1.”  where are the bars here ? 

p < 0.05 or  p> 0.05 which one is correct expression ? 

It is ok, however, some gramaical mistckes may corrected

Author Response

Comments to the authors
I have completed the evaluation of  manuscript entitled ” Novel single washing approach for eco-efficient production of  mackerel (Auxis thazard) surimi by the aid of ultrasonication”. authors demonstrated that Although the surimi yield was lower after washing with UW, but the  overall surimi quality was improved. Most important finding of this paper may be highlighted that this was aimed to reduce waste water production in surimi plants by washing with unltrasonication . 

Specific points are as below

It is was  felt that  a better title may be thought of to depict work done clearly.

Ans: The title was changed to “Single Ultrasonic-Assisted Washing for Eco-Efficient Production of Mackerel (Auxis thazard) Surimi”

It will be better that for treatments, A1, A2….so on shall be prepared in a table for clear understanding.

Ans: It was given in Table 1. Washing process used in each treatment.

The amplitude of the ultrasonication could not found anywhere inn manuscript ? Futher , how did you decide the amplitude for the untrasonication ?

Ans: The detail about the ultrasonication was given. “An ultrasound treatment was carried out using an ultrasonicator (Sonics, Model VC750, Sonica & Materials, Inc., Newtown, USA). A flat-tip probe with a diameter of 25 mm was employed with an intensity of 153 W/cm2 at a single frequency of 20 kHz and an amplitude of 80%.”

2.4.6 Evaluation of Rancid and Fishy Odors- Ten participants (20–40 years old) tested the gel's sensory properties …..why only ten  participants. It should have been atleast 20 ?

Ans: Thank you very much. Herein, we used trained panelists with extensive expertise in assessing rancid and fishy odors. It was originally stated that “The panelists were recruited for their expertise in assessing the flavor of cooked surimi gel as well as their sensitivity to fishy and rancid odors. They were all trained to have a thorough understanding of rancid and fishy odors [10].

The statement was also added “All of the panelists are researchers and graduate students who conduct research on fish and seafood with adequate skills in sensory evaluation of fish and surimi products.”

In the future, however, we will consider 20 or more panelists for sensory evaluation.

3.5 Rheological Property

Line 407-408 G' and G'' remained  constant during heating from 20 to 50 °C, then dropped slightly between 50 to 55 C, in indicating partial denaturation of myofibillar protein. Similar drop in G’ and G” also reported in squid (Mehta et al., 2017 (https://ifst.onlinelibrary.wiley.com/doi/abs/10.1111/jfpp.12891) . Where they have corroborated this drop or denaturation to catheptic activity also increment at this particular range. However, this range may vary species to species. That can help us to design better cooking temperature for the gel. Hope this is help for the authors. It is advisable to add above discussion and reference to improve the discussion in  this section.  

Ans: Thank you very much. The suggested reference was used for discussion. “G' changes during heating were caused by denaturation and cross-linking of various myofibrillar protein components at different temperatures. G' and G'' remained constant during heating from 20 to 50 °C, then dropped slightly between 50 to 55 °C, indicating partial denaturation of myofibillar protein as well as residual protease activity [40]. A similar drop in G' and G" was also observed in Indian squid (Loligo duvauceli) mantle proteins [40].”

Following that, the reference list was updated.

  1. Mehta, N.K.; Balange, A.K.; Lekshmi, M.; Nayak, B.B. Changes in dynamic viscoelastic and functional properties of Indian squid mantle during ice Storage. J. Food Process. Preserv. 2017, 41(3), e12891.

Figure 5. FTIR spectra of gels from mackerel unwashed mince and surimi prepared 604 using conventional washing and ultrasonic assisted single washing processes.

Bars  represent the standard deviations from triplicate determinations. Different letters 606 indicate the significant differences (< 0.05). Treatment caption can be seen in Table 1.”  where are the bars here ? 

< 0.05 or  p> 0.05 which one is correct expression ? 

Ans: We apologize for the error. Actually, the figure did not require the bars. It was thus removed.

Comments on the Quality of English Language

It is ok, however, some gramaical mistckes may corrected

Ans: Thank you very much. To double-check the English, QuillBot, a paraphrase tool, was used.

Reviewer 2 Report

Overview: The study delves into a promising single washing method for producing dark-fleshed mackerel surimi facilitated by ultrasonication alongside cold carbonated water with a mix of 0.6% NaCl and combined antioxidants. Various washing durations (5, 10, and 15 minutes) were evaluated both with and without the use of ultrasound. The findings suggest that surimi yield decreases as the ultrasonic treatment time lengthens.

Strengths:

  1. Methodological Approach: The study adopts a novel method, merging ultrasound with carbonated water and antioxidants, aiming to optimize surimi production.
  2. Comprehensive Comparison: The paper contrasts surimis produced under different washing and ultrasonic conditions, offering a robust analysis of the effects of these variables.
  3. Environmental Relevance: The exploration of reduced washing time and the consequent decrease in water usage is particularly pertinent in a sustainability context.

Weaknesses:

  1. Surimi Preparation: While the abstract provides insights into the treatments applied, the full description of surimi preparation in the main article might need further elaboration.
  2. Ultrasound Conditions: The abstract does not explicitly specify the ultrasonic conditions employed, such as frequency and intensity. This is crucial for experiment replicability.
  3. Treatment Table: A tabulated summary of the treatments (from A1 to A8) would aid reader comprehension, swiftly conveying the study's variations and comparisons.
  4. Analysis Repetitions: The number of repetitions conducted for each analysis is not mentioned, which is essential for gauging the robustness and reliability of the outcomes.

Recommendations for Improvement:

  1. Surimi Preparation Details: Expand on the surimi preparation section, offering more specifics about ingredients, ratios, methodologies, and equipment utilized.
  2. Specify Ultrasound Conditions: Incorporate detailed information on the ultrasonic conditions, such as frequency, intensity, and other relevant parameters.
  3. Include Treatment Table: Insert a table that clearly lists and describes the different treatments used in the study.
  4. Repetition Details: Specify the number of repetitions conducted for each analysis, and if multiple repetitions were done, also present the mean and standard deviation.

Conclusion: This research signifies a valuable contribution to the optimized production of dark-fleshed fish surimi. Addressing the above recommendations would significantly bolster the clarity and robustness of the presented research.

Author Response

Overview: The study delves into a promising single washing method for producing dark-fleshed mackerel surimi facilitated by ultrasonication alongside cold carbonated water with a mix of 0.6% NaCl and combined antioxidants. Various washing durations (5, 10, and 15 minutes) were evaluated both with and without the use of ultrasound. The findings suggest that surimi yield decreases as the ultrasonic treatment time lengthens.

Strengths:

  1. Methodological Approach: The study adopts a novel method, merging ultrasound with carbonated water and antioxidants, aiming to optimize surimi production.
  2. Comprehensive Comparison: The paper contrasts surimis produced under different washing and ultrasonic conditions, offering a robust analysis of the effects of these variables.
  3. Environmental Relevance: The exploration of reduced washing time and the consequent decrease in water usage is particularly pertinent in a sustainability context.

Ans: Thank you very much for the kind remarks about our manuscript.

Weaknesses:

  1. Surimi Preparation: While the abstract provides insights into the treatments applied, the full description of surimi preparation in the main article might need further elaboration.

Ans: The surimi production was detailed. “To prepare surimi, different washing periods (5, 10, and 15 min/cycle) with and without ultrasound were tested in this study using a single washing cycle of CW containing 0.6% NaCl and mixed antioxidant (0.2% sodium tripolyphosphate/0.2% sodium erythorbate/0.5% EDTA) (CSA) [10]. A medium to mince ratio of 3 to 1 (v/w) was used for washing. Unwashed mince (A1) and conventional water-washed surimi (10 min/cycle, 3 cycles) (A2) were used as controls. A3, A4, and A5 had ultrasound-assisted washing for 5, 10, and 15 min, respectively, whereas A6, A7, and A8 had non-ultrasound-assisted washing for 5, 10, and 15 min (Table 1). An ultrasound treatment was carried out using an ultrasonicator (Sonics, Model VC750, Sonica & Materials, Inc., Newtown, USA). A flat-tip probe with a diameter of 25 mm was employed with an intensity of 153 W/cm2 at a single frequency of 20 kHz and an amplitude of 80%. After being washed, mince was dewatered by being passed through a layer of nylon screen, then hydraulically pressed to achieve a final moisture content of about 80%. The yield of surimi from various washing practices was calculated based on the weight of surimi in relation to the weight of minced fish. Each sample was well combined with 4% (w/w) sucrose and 4% (w/w) sorbitol before being frozen in an air blast freezer (Polar DN494 367 Blast Freezer, Campbell town, NSW, Australia). The frozen samples were stored at -18 °C until they were analyzed. Surimi underwent extensive testing for physicochemical properties and gel-forming ability.

  1. Ultrasound Conditions: The abstract does not explicitly specify the ultrasonic conditions employed, such as frequency and intensity. This is crucial for experiment replicability.

Ans: The ultrasound condition was given in Section 2.2 “An ultrasound treatment was carried out using an ultrasonicator (Sonics, Model VC750, Sonica & Materials, Inc., Newtown, USA). A flat-tip probe with a diameter of 25 mm was employed with an intensity of 153 W/cm2 at a single frequency of 20 kHz and an amplitude of 80%.”

  1. Treatment Table: A tabulated summary of the treatments (from A1 to A8) would aid reader comprehension, swiftly conveying the study's variations and comparisons.

Ans: As also recommended by Reviewer#1, the treatment was presented in Table 1.

  1. Analysis Repetitions: The number of repetitions conducted for each analysis is not mentioned, which is essential for gauging the robustness and reliability of the outcomes.

Ans: In the Section 2.5 Statistical Analysis, the number of repetitions conducted for each analysis was mentioned. “All of the measurements were run in triplicate (n = 3), with the exception of the sensory assessment, which was obtained from a panel of 10 participants.”

Recommendations for Improvement:

  1. Surimi Preparation Details: Expand on the surimi preparation section, offering more specifics about ingredients, ratios, methodologies, and equipment utilized.

Ans: The surimi production was detailed. “To prepare surimi, different washing periods (5, 10, and 15 min/cycle) with and without ultrasound were tested in this study using a single washing cycle of CW containing 0.6% NaCl and mixed antioxidant (0.2% sodium tripolyphosphate/0.2% sodium erythorbate/0.5% EDTA) (CSA) [10]. A medium to mince ratio of 3 to 1 (v/w) was used for washing. Unwashed mince (A1) and conventional water-washed surimi (10 min/cycle, 3 cycles) (A2) were used as controls. A3, A4, and A5 had ultrasound-assisted washing for 5, 10, and 15 min, respectively, whereas A6, A7, and A8 had non-ultrasound-assisted washing for 5, 10, and 15 min (Table 1). An ultrasound treatment was carried out using an ultrasonicator (Sonics, Model VC750, Sonica & Materials, Inc., Newtown, USA). A flat-tip probe with a diameter of 25 mm was employed with an intensity of 153 W/cm2 at a single frequency of 20 kHz and an amplitude of 80%. After being washed, mince was dewatered by being passed through a layer of nylon screen, then hydraulically pressed to achieve a final moisture content of about 80%. The yield of surimi from various washing practices was calculated based on the weight of surimi in relation to the weight of minced fish. Each sample was well combined with 4% (w/w) sucrose and 4% (w/w) sorbitol before being frozen in an air blast freezer (Polar DN494 367 Blast Freezer, Campbell town, NSW, Australia). The frozen samples were stored at -18 °C until they were analyzed. Surimi underwent extensive testing for physicochemical properties and gel-forming ability.

  1. Specify Ultrasound Conditions: Incorporate detailed information on the ultrasonic conditions, such as frequency, intensity, and other relevant parameters.

Ans: The ultrasound condition was given in Section 2.2 “An ultrasound treatment was carried out using an ultrasonicator (Sonics, Model VC750, Sonica & Materials, Inc., Newtown, USA). A flat-tip probe with a diameter of 25 mm was employed with an intensity of 153 W/cm2 at a single frequency of 20 kHz and an amplitude of 80%.”

  1. Include Treatment Table: Insert a table that clearly lists and describes the different treatments used in the study.

Ans: As also recommended by Reviewer#1, the treatment was presented in Table 1.

  1. Repetition Details: Specify the number of repetitions conducted for each analysis, and if multiple repetitions were done, also present the mean and standard deviation.

Ans: In the Section 2.5 Statistical Analysis, the number of repetitions conducted for each analysis was mentioned. “All of the measurements were run in triplicate (n = 3), with the exception of the sensory assessment, which was obtained from a panel of 10 participants.”

Conclusion: This research signifies a valuable contribution to the optimized production of dark-fleshed fish surimi. Addressing the above recommendations would significantly bolster the clarity and robustness of the presented research.

Ans: Thank you very much. All of the points raised above were addressed. A conclusion was also rewritten. “A single washing assisted by ultrasonication (UW) in conjunction with carbonated water containing salt and mixed antioxidants (CSA) was an innovative process for producing mackerel surimi. Although the surimi yield was lower after washing with UW, the removal of unwanted proteins and overall surimi quality were improved. A longer UW treatment time resulted in more protein denaturation, protein oxidation, and lipid oxidation in surimi. The surimi made by CSA-UW for 5 min (A3) had comparable or even better overall surimi quality than 3-cycle washed surimi. The correspondence gel (A3) exhibited comparable gel strength and whiteness, as well as a high water holding capacity, to the control gel (A2). This was supported by the formation of regularly continuous and smooth network structures in surimi gel. Thus, a single CSA-UW washing for 5 min could potentially prepare the same quality surimi from dark-fleshed fish muscle as a 3-cycle conventional washing method. This could lay the groundwork for the development of surimi made from dark-fleshed fish, which would require less washing time and produce less waste water.

Reviewer 3 Report

Novel single washing approach for eco-efficient production of 2 mackerel (Auxis thazard) surimi by the aid of ultrasonication is a research article holding a manuscript ID 2641826 is the study about the surimi production through the use of ultrasound and CSA water with an aim to reduce washing time and washing water. This is an interesting study, holding a possibility for commercial applications. The manuscript is well written with detailed explanation and discussion of the results.

One major question – In abstract and conclusion, the authors claimed that the FTIR study showed “the functional groups of protein were similar” (line 27), while “protein denaturation, protein oxidation, myoglobin removal, and lipid oxidation” were also claimed (line 20-21), which seems contradict to each other. Please explain!

Minor issues:

Line 14-15: (1.5 mM EDTA/0.2% sodium erythorbate/0.2% sodium tripolyphosphate) – same unit for all three chemicals would be easy for readers to follow the method?

Line 70: unpresent compound?

Line 84: analyses?

Line 104: “4% sucrose and 4% sorbitol” – please mention w/w or w/v or v/w or v/v.

Line 127: “milligrams per gram of sample” – please express in the same as in section 2.3.3.

Line 134: Please provide the meaning of DeoMB, OxyMb, MetMb, A528, A525, A557, A503.

Line 148: extimated?

Line 177: elasti-ty?

Table 2: Please express in the same unit for myoglobin and its redox state (mg/g Vs. %)?

Line 369: fore?

Section 3.5: Please change to the subsection title as “Rheological Properties”

Line 609-615: The description is not related to Fig. 5; while the authors missed to mention Fig. 6 in the text?

Minor English grammar and typos corrections are recommended.

Author Response

Novel single washing approach for eco-efficient production of 2 mackerel (Auxis thazard) surimi by the aid of ultrasonication is a research article holding a manuscript ID 2641826 is the study about the surimi production through the use of ultrasound and CSA water with an aim to reduce washing time and washing water. This is an interesting study, holding a possibility for commercial applications. The manuscript is well written with detailed explanation and discussion of the results.

Ans: Thank you very much.

One major question – In abstract and conclusion, the authors claimed that the FTIR study showed “the functional groups of protein were similar” (line 27), while “protein denaturation, protein oxidation, myoglobin removal, and lipid oxidation” were also claimed (line 20-21), which seems contradict to each other. Please explain!

Ans: The sentence in lines 20–21 described the characteristics of surimi. “Increased ultrasonic time resulted in greater protein denaturation, protein oxidation, myoglobin removal, and lipid oxidation in surimi (p < 0.05).” whereas line 27 addressed the FTIR spectra of gel.

To clear up any uncertainty, the term "protein gel" was added to line 27, where the sentence related to the gel.  “All of the surimi gels had identical FTIR spectra, indicating that the functional groups of the protein gel were consistent throughout.

Minor issues:

Line 14-15: (1.5 mM EDTA/0.2% sodium erythorbate/0.2% sodium tripolyphosphate) – same unit for all three chemicals would be easy for readers to follow the method?

Ans: It was changed to “0.5% EDTA/0.2% sodium erythorbate/0.2% sodium tripolyphosphate.”

Line 70: unpresent compound?

Ans: It was changed to “unpleasant”.

Line 84: analyses?

Ans: It was changed to “analysis”.

Line 104: “4% sucrose and 4% sorbitol” – please mention w/w or w/v or v/w or v/v.

Ans: It was changed to “Each sample was well combined with 4% (w/w) sucrose and 4% (w/w) sorbitol before being frozen in an air blast freezer.” 

Line 127: “milligrams per gram of sample” – please express in the same as in section 2.3.3.

Ans: It was expressed as g/100 g sample as same as its redox state as suggested for Table 2.

Line 134: Please provide the meaning of DeoMB, OxyMb, MetMb, A528, A525, A557, A503.

Ans: Done.

Line 148: extimated?

Ans: It was changed to “estimated”.

Line 177: elasti-ty?

Ans: It was changed to “elasticity”.

Table 2: Please express in the same unit for myoglobin and its redox state (mg/g Vs. %)?

Ans: Myoglobin was measured as "g/100 g" whereas the redox state was measured as the ratio of three different derivatives and was given as %.

Line 369: fore?

Ans: It was changed to “force”.

Section 3.5: Please change to the subsection title as “Rheological Properties”

Ans: It was changed to “Rheological Properties”.

Line 609-615: The description is not related to Fig. 5; while the authors missed to mention Fig. 6 in the text?

Ans: I apologize sincerely for the error. Fig. 6 has already been addressed.

Comments on the Quality of English Language

Minor English grammar and typos corrections are recommended.

Ans: Thank you very much. To double-check the English, QuillBot, a paraphrase tool, was used.

Round 2

Reviewer 2 Report

The authors made the requested modifications.